# Biomass Resources of *Phragmites australis* in Kazakhstan: Historical Developments, Utilization, and Prospects

**Azim Baibagyssov** [1,2,3,*], **Niels Thevs** [2,4], **Sabir Nurtazin** [1], **Rainer Waldhardt** [3], **Volker Beckmann** [2]  **and Ruslan Salmurzauly** [1]

1 Faculty of Biology and Biotechnology, Al-Farabi Kazakh National University, Almaty 050010, Kazakhstan; nurtazin.sabir@gmail.com (S.N.); ruslaan.200587@gmail.com (R.S.)
2 Faculty of Law and Economics & Institute of Botany and Landscape Ecology, University of Greifswald, 17489 Greifswald, Germany; n.thevs@cgiar.org (N.T.); volker.beckmann@uni-greifswald.de (V.B.)
3 Division of Landscape Ecology and Landscape Planning, Institute of Landscape Ecology and Resources Management, Center for International Development and Environmental Research (ZEU), Justus Liebig University Giessen, 35390 Giessen, Germany; rainer.waldhardt@umwelt.uni-giessen.de
4 Central Asia Office, World Agroforestry Center, Bishkek 720001, Kyrgyzstan
* Correspondence: azim.baibagyssov@agrar.uni-giessen.de or azim.baibagysov@gmail.com

**Abstract:** Common reed (*Phragmites australis* (Cav.) Trin. Ex Steud.) is a highly productive wetland plant and a potentially valuable source of renewable biomass worldwide. There is more than 10 million ha of reed area globally, distributed mainly across Eurasia followed by America and Africa. The literature analysis in this paper revealed that Kazakhstan alone harbored ca. 1,600,000–3,000,000 ha of reed area, mostly distributed in the deltas and along the rivers of the country. Herein, we explored the total reed biomass stock of 17 million t year$^{-1}$ which is potentially available for harvesting in the context of wise use of wetlands. The aim of this paper is to reveal the distribution of reed resource potential in wetland areas of 13 provinces of Kazakhstan and the prospects for its sustainable utilization. Reed can be used as feedstock as an energy source for the production of pellets and biofuels, as lignocellulosic biomass for the production of high strength fibers for novel construction and packaging materials, and innovative polymers for lightweight engineering plastics and adhesive coatings. Thereby, it is unlikely that reed competes for land that otherwise is used for food production.

**Keywords:** reed beds; wetlands; bioeconomy; feedstock; Central Asia; utilization; Soviet Socialist Republics

## 1. Introduction

Oil, gas, and coal deposits are limited fossil resources, and their combustion contributes to climate change by releasing large amounts of carbon dioxide into the atmosphere. Finiteness of those resources and necessity to reduce the $CO_2$ burden of the atmosphere significantly increase the importance of biomass as a resource for energy production or as a chemical feedstock for innovative utilization pathways within knowledge-based production [1,2]. This shift is often called bioeconomy, and its growth augments the use of biomass as a primary renewable feedstock for material and energy use [3–5]. Increasing the supply of biomass, however, often results in competition with food and feed production, and leads, in the short or long term, to food-feed-fuel conflicts, for example, in the case of increased utilization of corn or other food crops as feedstock for biogas or biofuels [6–8]. To minimize these aforesaid food-feed-fuel conflicts, sites unsuitable for regular farming can be utilized for biomass production. Significant potential against this background is seen in the conservation and wise use of

wetlands and reed bed areas [9–11]. While their sustainable management for benefits of the exploitation of reed has become increasingly attractive from the economic perspective [12,13], the growing demand for sustainable biomass portrays wetlands and reed bed areas as increasingly interesting for a circular bioeconomy. In this context, widespread wetland plant common reed (*Phragmites australis*) has become an increasingly attractive source of renewable biomass from its monotonous reed beds. Next to their high productivity, they also provide a broad range of ecosystem services (e.g., resources provision, recreation, flood mitigation, and aesthetics) and a habitat of wildlife [14–17].

Globally, there are more than 10 million ha of reed bed area, as calculated by [18] and summarized in a review study [19], mostly spread across Eurasia followed by America [20]. About half of the total area exists in the former Soviet Union [21–23], shared mainly by Kazakhstan and other Central Asian countries [24,25].

For a couple of decades, reed beds have been neglected in the research, regardless of this fact, Kazakhstan harbors about two million ha and, more importantly, it is the country with one of the most substantial reed bed areas worldwide according to [26] cited in [27]. While the spatial distribution and assessed area of wetlands and reed biomass resources have been previously demonstrated for the Ili Delta in southeast Kazakhstan [28], to date, a limited attempt has been made to conduct any assessment across the country. Against this background, this study focused on the type of information that is available about the number of reed resources, its development, and spatial distribution. The purpose of this study was to determine the resource potential and biomass perspectives of common reed (*Phragmites australis*) in Kazakhstan. The research looks at the driving forces of past, current, and future developments of reed resources and how they influence the utilization of reed resources.

The study was motivated by three main questions as follows:

- What is the current development and state of reed resources in Kazakhstan?
- What are the driving forces of wetlands and reed bed areas change in Kazakhstan?
- What past, current, and future utilization of reed resources are documented and possible in the future?

In addressing these research questions, detailed data on reed bed areas and visual illustration of their spatial distribution in the 13 provinces of the country are provided. Moreover, the drivers affecting the development of wetland areas across Kazakhstan and a showcase of the largest in Central Asia, the Ili Delta, as a wetland complex lying downstream in a transboundary river basin is provided with retrospective analyses in the changes of wetlands and reed bed areas in the course of the last 50 years. The research was conducted using the narrative literature review method based on the international and, in particular, the Russian and Kazakh literature. The authors used archive materials and reports available for the period of study, publications, dissertations, several internationally funded project reports, and their findings from investigations on the experience of projects fulfilled in Ili Delta, Balkhash Lake, Kazakhstan. This publication improves our knowledge about wetlands and reed bed areas in dryland regions and their importance as a core from which a bioeconomy can obtain its feedstock. Given the limited information on reed biomass resources in a drylans country such as a Kazakhstan, this paper also contributes towards advancing the knowledge on the current assessment of reed biomass resources and their potential as a valuable source of renewable biomass for a bioeconomy in Kazakhstan and beyond.

In an attempt to present the resource potential and biomass perspectives of *Phragmites australis* in Kazakhstan, we developed a structure for the review, based on literature related to reed, its resources, and distribution in wetland areas of Kazakhstan. The review consists of four sections. In Section 1, we highlight the distribution of common reed and its biomass potential globally with the share of Kazakhstan. In Section 2, we identify the current reed resources based on the most recent available data on reed bed distribution in wetland areas of Kazakhstan with an emphasis on a historical overview of reed resources in the largest wetland complex in the region, i.e., Ili Delta. In Section 3, we specify the driving forces of change in the wetlands and reed bed areas in Kazakhstan. In Section 4, the review

discerns past, current, and prospect developments of reed resources and their utilization in Kazakhstan. Lastly, in the Discussion and Conclusion section we identify the limitation of the study and summarize the outcome information referred to research questions.

## 2. Reed Ecology and Share of Kazakhstan in the Global Reed Biomass Potential

### 2.1. Ecology and Distribution of the Common Reed

Common reed (*Phragmites australis* (Cav.) Trin. Ex Steud.) is one of the widespread wetland plants worldwide. Europe, the Middle East, and America are the core distribution areas, according to Haslam [20]. It grows mostly in wetlands on the shores of lakes and gulfs, along riverbanks, and on nutrient-rich peatlands. Usually, it forms large monospecies stands in virtue of the monocultural nature and it has a tendency to propagate vegetatively. The preferred water level ranges from slightly below the soil surface to one meter above ground level. However, on the one hand, it can also grow in deep water, with some cases up to about four meters in highly transparent oligotrophic waters. On the other hand, reed also grows in soils with groundwater levels of one to two meters below the surface [29,30]. It forms small stands at desert margins with water tables of around four meters below the ground surface. Although reed mainly grows in fresh water, it can also grow in up to 16% salt water, which exhibits its tolerance or adaptability potential to these scenarios [31,32].

Only reed rhizomes are perennial and, in colder climates, the aboveground part of the plant dies after the growing season [33]. Nutrients are relocated from the stem and leaves down to the rhizomes and stored there for the next growing season [34]. Although reed can germinate from seeds, sexual reproduction takes place to a limited extent [35]. In contrast, its reproduction in a vegetative way is much more prevalent [36]. Having a strong ability to propagate from rhizomes, it forms a dense, impenetrable layer where very few other species can compete. Therefore, in areas where reed settles as a pioneer plant, it forms a stable monospecific stand with its absolute predominance [36,37]. As *P. australis* can spread very rapidly into new areas and persist there, it is seen as an invasive species harmful to both species diversity and wildlife habitat in some areas around the world (e.g., North America and Oceania) [38–40].

### 2.2. Global Reed Biomass Potential and Share of Kazakhstan

Reed is a tall, highly productive perennial sweet grass according to [13,19]. Although aboveground net primary production varies from less than three $t\ ha^{-1}\ year^{-1}$ to as much as 30 $t\ ha^{-1}\ year^{-1}$ [41], the highest yields of 60 $t\ ha^{-1}\ year^{-1}$ has been observed on the coast of the Red Sea [13]. As a cosmopolitan vegetation, it occurs all over the world (except in Antarctica) with the global area more than 10 million ha, as summarized in Table 1 by [19]. Having a minimal annual harvested yield of three $t\ ha^{-1}$ of dry biomass, reed has a theoretical global biomass potential of at least 30 million $t\ year^{-1}$.

**Table 1.** Reed bed areas and aboveground biomass in different countries, updated with the most recent available data on reed beds area after [19].

| Site/Region/Country | Reed Beds Area (ha) | Average Productivity ($t\ ha^{-1}\ year^{-1}$) | Total Biomass ($t\ year^{-1}$) |
|---|---|---|---|
| Former USSR [1] (Union of Soviet Socialist Republics) | | | |
| Estonia | 27,899 (12,970 harvestable) | 6.8 | 88,368 |
| Only lakes, Latvia | 13,400 (10,826 harvestable) | 7.2 | 97,000 |
| Curonian Lagoon, Lithuania | 4995 | - | - |
| Kaliningrad Oblast, Russia | 200–300 | - | - |
| Danube Delta, Ukraine | 105,055 | 5 | 50,000 |
| Regions and provinces of Russia | >1,715,000 | - | - |
| Kazakhstan | 1,600,000–3,000,000 | 8.2 | 17,000,000 |
| Uzbekistan | 1,560,000 (372,630 harvestable) | 16 | 5,961,960 |
| Turkmenistan | 1,000,000 | - | - |

**Table 1.** *Cont.*

| Site/Region/Country | Reed Beds Area (ha) | Average Productivity (t ha$^{-1}$ year$^{-1}$) | Total Biomass (t year$^{-1}$) |
|---|---|---|---|
| European Countries | | | |
| Poland | 60,000 | - | - |
| South Finland | 30,000 (15,000 harvestable) | 10 | 150,000 |
| South Sweden | 230,000 | 5 | 1,150,000 |
| Mecklenburg-Vorpommern, Germany | 1500 | | |
| The Netherlands | 9000 (2850 harvested) | - | - |
| Lake Neusiedl, Austria | 60,000 (36,000 harvestable) | 7 | 28,500 |
| United Kingdom | 7700 managed for conservation | - | - |
| Danube Delta, Romania | 200,000 (125,000 harvested) | 26 | 3,250,000 |
| Hungary | 26,200 | - | - |
| America | | | |
| Brackish, salt and tidal marshes, USA | 1,800,000 | - | - |
| Asia | | | |
| NW, N, NE, and coastal east China [2] | 1,005,000 (484,000 harvested) | 5.5 | 5,527,500 |
| North and South Korea | 30,000 and 20,000 | - | - |
| Iraq | 17,300 | - | - |
| Globally | >10,000,000 | - | >30,000,000 |

[1] The total area of the former USSR was 5,500,000 ha. [2] Reed covered an area of 1 million ha (484,000 ha of planted reed) outside of protected areas in China, in 2004.

## 3. Wetlands and Reed Bed Areas Distribution in Kazakhstan

### 3.1. Wetlands, Reed Beds, and the Transboundary Water Situation in Kazakhstan

Kazakhstan with a total area of just over 2.72 million km$^2$ is the world's largest landlocked country. Located in Central Asia, it is covered mainly by drylands. Due to its geographical location, Kazakhstan has a sharp continental arid to semiarid climate, which is characterized by a strong north-south gradient in temperature and precipitation and a gradient from lowlands to high mountain areas [42]. More than 80% of the total country's area account as steppes and temperate (or winter-cold) deserts with precipitation of 150–200 mm year$^{-1}$ in the center and the south; only in the northern part of Kazakhstan precipitation exceeds 300 mm year$^{-1}$ [43,44]. Average annual rainfall in mountain areas can reach 880 mm.

The region of Central Asia is home to many large rivers surrounded by endorheic (i.e., missing an outlet to the ocean) basins with several thousand lakes. As the main lifeline in drylands, they form essential habitats such as wetlands with the vast areas of reed beds. Kazakhstan alone has more than 3000 natural lakes with a surface area larger than 1 km$^2$ and more than 200 water reservoirs [45]. The major rivers in Kazakhstan, i.e., Syr Darya, Ili, Irtysh, Chu, and the Ural, all originate in mountain areas of one or more neighboring countries but have their middle and lower reaches in Kazakhstan and eventually drain into inland deltas or end lakes also in Kazakhstan. Many of the small lakes are part of small river basins that are not shared with other countries. Many of those small lakes are not perennial, as they depend on the autumn and spring precipitation and dry out during the hot and dry summers.

Under the semiarid and arid conditions in most parts of Kazakhstan, those major and transboundary rivers are of high economic importance. As in neighboring Central Asian countries, a significant part of the population inhabits the basins of those rivers and is engaged in irrigated farming, which practically produces the total agricultural output [46,47]. However, there is a competitive demand for water among countries and regions in the country due to the existing uneven distribution of water resources, which has been resulting in the current upstream-downstream conflicts over the past three decades [48].

Consequently, today these conflicts over water pose a high risk to the economies of the region, to regional cooperation, and security [49,50]. In this regard, Kazakhstan, being downstream, is more

vulnerable to future water shortages than neighboring upstream countries. In addition to the competition between upstream and downstream countries and regions, there is also competition for water between various water users within the country, such as natural ecosystems and irrigated agriculture [51,52]. This is amplified against the background of ever-increasing water consumption due to a reasonably high population growth rate. The depletion of water sources and their pollution, along with climate change, contribute to further deterioration of sanitary and environmental conditions.

## 3.2. Historical Distribution of Reed Bed Areas in Kazakhstan

One of the first to publish data on areas under reed beds was the Institute of Economics of the Academy of Sciences of the Kazakh SSR (Soviet Socialistic Republic), in 1962. Reed beds in the country accounted for about 1,042,000 ha (Table 2).

**Table 2.** Reed bed areas distribution in different provinces of Kazakh SSR (source [53], 1958 cited in [54], 1964).

| Province Names * | Reed Beds Area (ha) | Average Productivity (t ha⁻¹ year⁻¹) | Total Biomass (t year⁻¹) | Largest Reed Bed Sites |
|---|---|---|---|---|
| Alma-Ata | 250,000 | 15 | 3,750,000 | Ili River and tributaries of Lake Balkhash |
| Guryev | 170,000 | 15 | 2,550,000 | The northern coast of the Caspian Sea, Rivers Emba and the Ural |
| Kzyl-Orda | 126,000 | 15 | 1,890,000 | Syr Darya River, Lake Aschikul |
| Dzhambul | 97,000 | 15 | 1,455,000 | Rivers Chu, Talas and Kuragaty, Lake Balkhash |
| Chimkent | 73,000 | 15 | 1,095,000 | Rivers Syr Darya and Chu |
| Taldy-Kurgan | 65,000 | 15 | 975,000 | Ili River, Lakes Alakul and Sasykkul |
| Kustanai | 58,000 | 12 | 696,000 | Turgai River and Lakes Sarymoni, Zharkul, Sarykopa, and Kamyshlykul |
| Aktyubinsk | 55,000 | 15 | 2,550,000 | Rivers Temir, Emba and Big Hobda |
| Uralsk | 40,000 | 10 | 400,000 | - |
| Semipalatinsk | 40,000 | 12 | 480,000 | Irtysh River, Lakes Alakul and Sasykkul |
| East Kazakhstan | 22,000 | 10 | 220,000 | Irtysh River and Lake Zaysan |
| Tselinograd | 15,000 | 10 | 150,000 | Lake Tengiz, Ishim River and its tributaries |
| Kokchetav | 10,000 | 8 | 50,000 | - |
| Pavlodar | 10,000 | 8 | 80,000 | Irtysh River |
| Karaganda | 6000 | 8 | 48,000 | Taldy River |
| North Kazakhstan | 5000 | 8 | 40,000 | Ishim River |
| Total | 1,042,000 | 14 | 14,679,000 | |

\* The names of the provinces are given for the administrative-territorial division of Kazakh SSR until 1962.

However, this area was corrected by [55] specifying it as the main reed beds of industrial importance and recorded according to the data of the Institute of Economics, as well as the Land Management Department of Regional Agricultural Administrations, the temporary scientific and technical commission of the State Scientific and Technical Committee of the Council of Ministers of the USSR, and the Institute of Botany of the Academy of Sciences of the Republic of Kazakhstan in 1964.

Moreover, according to [54] the Institute of Economics underestimated the reed bed area, since reed beds were taken into account only in 66 out of 192 administrative districts and only high-yielding reed beds available for industrial use were considered (See Appendix A for more detailed information). Hereupon, [54] included all provinces, as well as reed beds beyond the high yielding ones. Areas, productivity, and biomass stocks are given in Table 3. The estimated dry biomass stock for the whole Kazakh SSR was nearly 17 million t.

Figure 1 shows the reed bed distribution across Kazakhstan, as of 1964. Two-thirds of all reed beds are located in the southern part as follows: in the basins of the Syr Darya, Shu, and Ili Rivers, and on the coast of the Caspian Sea, including the Ural Delta.

**Table 3.** Reed bed areas, productivity, and reed biomass stocks in various provinces of Kazakh SSR (source [54], 1964).

| Province Names | Reed Beds Area (ha) | | Average Productivity (t ha⁻¹ year⁻¹) | Biomass Stocks (t) | |
|---|---|---|---|---|---|
| | Total | Including Accounted | | Total | Including Accounted |
| Alma-Ata * | 500,000 | 457,658 | 11 | 5,500,000 | 5,033,000 |
| Kzyl-Orda | 400,000 | 84,990 | 11 | 4,400,000 | 935,000 |
| Guryev | 275,000 | - | 8 | 2,200,000 | - |
| Dzambul | 100,000 | 88,355 | 12 | 1,200,000 | 1,059,000 |
| Chimkent | 100,000 | 87,540 | 8 | 800,000 | 700,000 |
| Kustanai | 85,000 | 80,802 | 6 | 510,000 | 480,000 |
| Aktyubinsk | 60,000 | 48,733 | 6 | 360,000 | 292,000 |
| Kokchetav | 60,000 | 56,079 | 6 | 360,000 | 336,000 |
| Semipalatinsk | 50,000 | 46,400 | 8 | 400,000 | 371,000 |
| Tselinograd | 50,000 | 38,400 | 8 | 400,000 | 307,000 |
| East Kazakhstan | 40,000 | 31,218 | 10 | 400,000 | 312,000 |
| North Kazakhstan | 35,000 | 29,698 | 5 | 175,000 | 134,000 |
| Uralsk | 15,000 | 13,400 | 8 | 120,000 | 108,000 |
| Pavlodar | 15,000 | 11,971 | 5 | 75,000 | 60,000 |
| Karaganda | 15,000 | 13,430 | 6 | 75,000 | 67,000 |
| Total | 1,800,000 | 1,088,674 | - | 16,975,000 | 10,194,000 |

* Due to territorial rearranges the Alma-Ata Province unites the so-called Alma-Ata and Taldy-Kurgan Provinces.

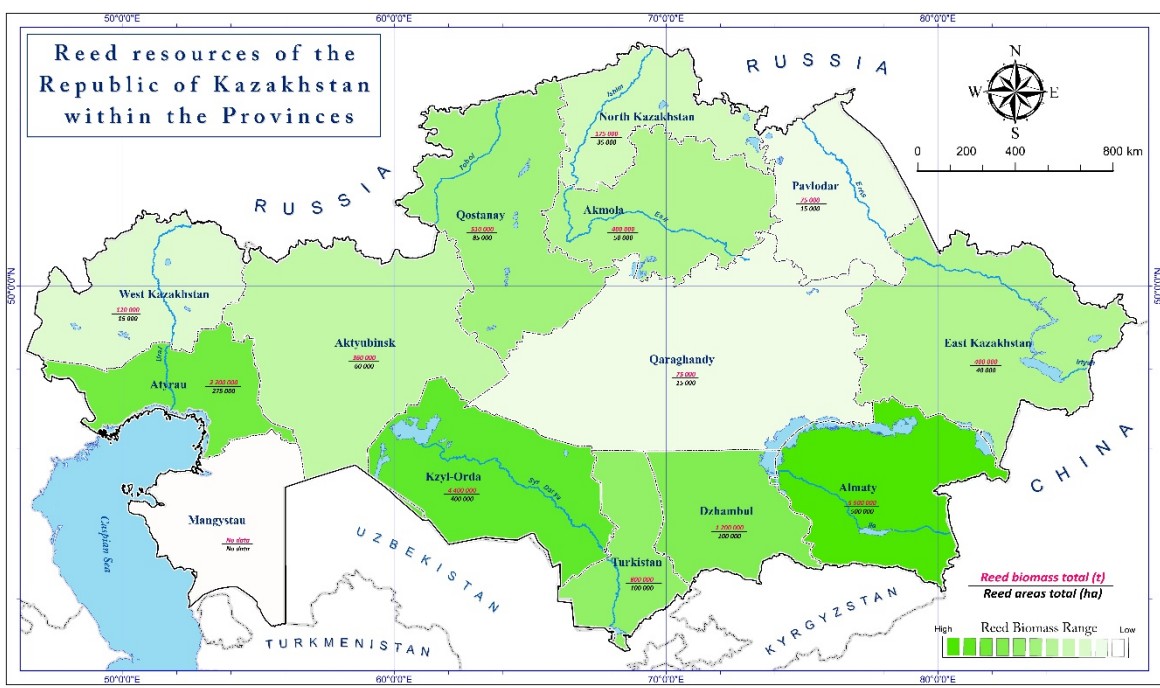

**Figure 1.** Map of reed resources within the provinces of Kazakhstan in white (no data) and gradient green color for reed biomass range between light green (75,000 t) and intense green (5.5 million t). (Authors, December 2019 based on [54], 1964).

Studies, conducted in 1979–1993, showed that the area occupied by high-yielding reed beds of industrial significance decreased by 70–95% in the Kzyl-Orda, Dzhambyl, Chimkent, and Alma-Ata Provinces as compared with the data of 1960 (Table 4), which are the provinces where the Syr Darya, Shu, and Ili River Basins are located. Reed beds in other provinces shrunk by 20–45% in areas and by 40–50% in terms of yields as compared to 1960.

**Table 4.** Changes in high-yielding reed beds of industrial significance in Kazakhstan from 1960 to 1990/1993 (source [55]).

| Province Names | Until 1960 | | | 1990–1993 | | |
|---|---|---|---|---|---|---|
| | Reed Beds Area (ha) | Average Productivity (t ha$^{-1}$ year$^{-1}$) | Production Stock (t) | Reed Beds Area (ha) | Average Productivity (t ha$^{-1}$ year$^{-1}$) | Production Stock (t) |
| Alma-Ata | 250,000 | 15 | 3,750,000 | 26,400 | 8.0 | 180,000 |
| Guryev | 170,000 | 15 | 2,550,000 | 51,000 | 8.5 | 433,500 |
| Kzyl-Orda | 126,000 | 15 | 1,890,000 | 11,800 | 8.6 | 86,600 |
| Dzambul | 97,000 | 15 | 1,455,000 | 8700 | 6.5 | 64,500 |
| Chimkent | 73,000 | 15 | 1,095,000 | 7200 | 8.6 | 63,000 |
| Taldy-Kurgan | 65,000 | 15 | 975,000 | 18,100 | 8.5 | 154,000 |
| Kustanai | 58,000 | 12 | 696,000 | 31,600 | 10.0 | 316,000 |
| Aktyubinsk | 55,000 | 14 | 770,000 | 33,500 | 10.0 | 335,000 |
| Uralsk | 40,000 | 10 | 400,000 | 32,000 | 10.0 | 320,000 |
| Semipalatinsk | 40,000 | 12 | 480,000 | 15,000 | 8.5 | 127,500 |
| East Kazakhstan | 22,000 | 10 | 220,000 | 13,200 | 8.5 | 82,200 |
| Tselinograd | 15,000 | 10 | 150,000 | 7500 | 8.0 | 61,000 |
| Kokchetav | 10,000 | 8 | 80,000 | 6000 | 7.0 | 42,000 |
| Pavlodar | 10,000 | 8 | 80,000 | 6000 | 7.0 | 42,000 |
| Karaganda | 6000 | 8 | 48,000 | 3500 | 7.0 | 24,500 |
| North Kazakhstan | 5000 | 8 | 40,000 | 3000 | 7.0 | 21,000 |
| Total | 1,042,000 | - | 14,679,000 | 274,300 | - | 2,352,700 |

*3.3. A Showcase of a Historical Overview of Reed Resources in the Ili Delta*

The Ili Delta is the only reed bed area, where the size and distribution of wetlands and reed bed areas were studied several times between the 1960s and today. These studies were done in the context of construction and filling of the Kapchagay Reservoir in the 1970s and 1980s, which is upstream of the Ili Delta. Later, studies were done, as the Ili Delta became one of the largest rather undisturbed wetland complexes after the Aral Sea had desiccated and adjacent deltas had shrunk substantially [56].

These studies tracked the changes of reed bed areas and related them to external factors, such as climate and anthropogenic impacts, which eventually informed how other reed beds across Kazakhstan could have changed from the 1960s until today.

Table 5 shows that the reed bed area in the Ili Delta dropped sharply from the 1960s until the beginning of the 1990s, but has grown substantially until today. The former is clearly attributed to the construction and filling of the Kapchagay Reservoir (see [56] and further literature there) and the creation of the Akdala irrigation unit. After the Kapchagay Reservoir was not further filled, more water was released downstream into the Ili Delta, which caused reed beds to increase again. This process was presumably supported by enhanced glacier melt in the headwaters and increased river runoffs as a result, which would be in line with a general trend of enhanced glacier melt in the course of climate change [57].

**Table 5.** Changes in high-yielding reed beds of industrial significance in the Ili Delta from 1960 to 2014/2015.

| Years | Total Reed Beds Area (ha) | Area of Production Reed Beds (ha) | Average Productivity (t ha$^{-1}$ year$^{-1}$) | Total Biomass (t year$^{-1}$) | Reference |
|---|---|---|---|---|---|
| Until 1960 | 363,200 | 142,700 | 15 | 5,448,000 | Demidovskaya [54] |
| 1990–1993 | 38,354 | 15,069 | 8 | 306,832 | Isambaev [55] |
| 2014–2015 | 211,778 | 85,400 | 10.2 | 2,155,218 | Thevs et al. [28] |

The numbers in Table 5 are in line with a change detection of vegetation indices from Landsat satellite images spanning from 1973 to 2013 [58]. The change detection revealed large areas of decreasing vegetation index from 1973–1990 and large areas of increasing vegetation index after 1990.

The time period after 2000, with regard to wetland distribution in the Ili Delta and other Ramsar Sites in Kazakhstan, was covered by Tesch and Thevs [59]. From 2000 to 2018, the reed bed area had not changed or slightly increased in most Ramsar Sites, also in the Ili Delta. Despite this general

trend, there were fluctuations in reed bed areas within this time span. In the Ili Delta, after a period of increasing reed bed area until 2010, the area decreased from 2010 to 2015 but increased again after 2015.

After Kazakhstan's independence, agriculture and irrigation receded, and therefore more water became available for natural riparian ecosystems in many river basins. Extrapolating these findings from the Ili Delta to other wetlands and reed bed areas of Kazakhstan, it can be concluded that reed bed areas shrank from the 1960s until the 1990s, but afterwards increased their areas again, although not reaching the sizes of the 1960s.

## 4. Driving Forces of Wetlands and Reed Bed Areas Change in Kazakhstan

### 4.1. Drivers of Wetlands and Reed Bed Areas Loss and Degradation

The most significant drivers of wetlands' change in Kazakhstan are mainly anthropogenic activities linked to the forms and scale of land use and water allocation in the basins of major rivers. Due to arid and semiarid climatic conditions, agriculture development in Kazakhstan and other Central Asian countries largely depends on water supply from mostly big transboundary rivers. Deliberate diversion of their water from its ordinary course with further stream diversions and withdrawals is the most pervasive way for irrigation. Although it contributes to the crop irrigation and land-use intensification, however, it often alters water regimes of wetlands and reed beds causing water levels to recede drastically.

According to Isambaev [55], regulation of the runoff of the Syr Darya, Talas, Chu, Ili, and Karatal Rivers after 1960 with the expansion of irrigated agriculture contributed to the absence of spring-summer floods along the rivers and lead to dramatic shrinking of reed bed areas (see Table 4). Thus, intensification of land use along major rivers and mainly in their upper catchment area resulted in water scarcity in the dry areas at the lower reaches, which are especially sensitive to water fluctuations. Furthermore, it resulted in ecological problems, such as land and vegetation degradation and biodiversity loss, as recent studies have revealed dramatic environmental, morphological, and areal changes of different Central Asian inland water bodies and lakes in the past decades [45,60–62]. Under these circumstances, terminal lakes and wetlands with reed bed areas in the semi-desert zone of Kazakhstan are regarded threatened due to an intensification of water use.

Fire is another significant threat for the reed beds that are located on non-submerged sites, meadow vegetation, parts of the Tugai forests, and shrub vegetation and their associated fauna. Fires are ignited by local people mainly in reed beds and partly other vegetation types to remove dead biomass, and thus enhance the productivity of reed to use it as pasture or hay meadow. Traditionally, such fires were most often ignited in winter or early spring, and usually with control for the least threat to fauna. However, today some fires are also ignited in summer and often get out of control [63,64]. Such uncontrolled fires destroy the habitat of wildlife (mainly mammals and birds) and decrease the productivity of the vegetation.

### 4.2. Drivers of Wetlands and Reed Beds Area Protection and Wise Use

Wetlands across Central Asia have been experiencing long-term degradation and are among the regions which are most severely affected by land degradation [59]. In particular, during the 20th and early 21st centuries, the rate of wetlands degradation dramatically increased due to human intervention. The most prominent example is the Aral Sea with the deltas of its main tributaries, the Amu Darya in Uzbekistan and the Syr Darya in Kazakhstan. Their desiccation has resulted from increased water diversion further upstream for irrigated agriculture. Environmental changes are observed in all-natural climatic zones, gradually intensifying downstream in the rivers and most significantly in the zone of runoff consumption. Subsequently, the Tengiz-Korgalzhyn Lakes System and Lakes in the Lower Irgiz and Turgai Rivers were the first sites located on the territory of the former Kazakh SSR that have been simultaneously listed on the Ramsar list by the Soviet Union Government, when it joined that convention in 1975 [65]. Both wetlands were relisted after the ratification of the

Convention by Kazakhstan as an independent state. The first site was confirmed in 2007 and the second separately in September 2011. Currently, the list of Ramsar sites in Kazakhstan has been increased to ten sites (Table 6) that have been declared as wetlands of international importance, with a total area of 3,281,398 ha [66].

**Table 6.** Wetland and reed bed sites of Kazakhstan recorded into the Ramsar list (source [66]).

| № | Name | Date of Record | Province | Area of the Ramsar Site (ha) | Protected Areas |
|---|------|----------------|----------|------------------------------|-----------------|
| 1 | Tengiz-Korgalzhyn Lake System | 11 October 1976 | Akmola Province | 353,341 | Korgalzhyn State Nature Reserve |
| 2 | Lakes of the lower Turgay and Irgiz | 11 October 1976 | Aktobe Province | 348,000 | Irgiz-Turgai Reserve, Turgai Nature Sanctuary of Republican Importance |
| 3 | Ural River Delta and the adjacent Caspian Sea coast | 10 March 2009 | Atyrau Province | 111,500 | "Akzhayik" Nature Reserve |
| 4 | Koibagar-Tyuntyugur Lake System | 7 May 2009 | Qostanai Province | 58,000 | - |
| 5 | Kulykol-Taldykol Lake System | 7 May 2009 | Qostanai Province | 8300 | - |
| 6 | Zharsor-Urkash Lake System | 12 July 2009 | Qostanai Province | 41,250 | Zharsor-Urkash Nature Sanctuary of Republican Importance |
| 7 | Naurzum Lake System | 12 July 2009 | Qostanai Province | 139,714 | Naurzum State Nature Reserve |
| 8 | Alakol-Sasykkol Lakes System | 25 November 2009 | Shared between Almaty and East Kazakhstan Provinces | 914,663 | Alakol State Nature Reserve |
| 9 | Ili River Delta and South Lake Balkhash | 1 January 2012 | Almaty Province | 976,630 | Balkhash, Karroy, and Kukan State Nature Sanctuaries of Republican Importance |
| 10 | Lesser Aral Sea and Delta of the Syrdarya River | 2 February 2012 | Kyzylorda Province | 330,000 | Barsakelmes State Nature Reserve |

This increase in the number of sites declared as wetlands of international importance is likely to have a positive effect on reed beds in the vision of importance for the conservation of global biological diversity and for sustaining human well-being through the maintenance of their ecosystem components, processes, benefits, and services. Consequently, in the framework of wise use, reed beds as productive ecosystems, concentrated mainly in wetlands, can be of economic importance for sustainable rural livelihoods. Thus, prudent management of these ecosystems with elements of sustainable harvesting of reed biomass can contribute to the general conservation of reed areas and the care of wetlands as an additional source of income for the local population.

## 5. Past, Current, and Prospect Developments of Reed Resources and Their Utilization in Kazakhstan

*5.1. Reed Resources Development and Utilization in the Past*

Although reed has been known to man since prehistoric times as a raw material supporting livelihoods [13,19,20], information on the use of reed in the territory of modern Kazakhstan up to

1920 is limited. The earliest notes on reed beds, however, were made by A.I. Maksheev during the expedition of A.I. Butakov to explore the Aral Sea, the Amu Darya, and Syr Darya Rivers from 1848 to 1850 [67]. Dense stands of tall grass and reeds along with riparian forest, the so-called "Tugai Forests", were identified as a unique habitat for tigers, which lived along floodplains of Central Asian rivers at that time [63,68,69]. Later, from 1850 to 1917, general information about the growth and use of reed in livelihoods were given in botanical works from the Tsarist time [70–74]. However, as these expeditions and research works covered a wide range of topics, there is limited information about the reed bed areas, their types, ecology, and productivity of reed in those sources.

Later, after Kazakhstan underwent significant socioeconomic and political developments that followed the establishment of the Soviet Union [75,76], reed became more and more interesting as a low-cost plant feedstock for paper and reed slabs. After a series of biochemical studies from 1930 to 1950 [55,77,78], in this regard, reed processing plants were established in the Kazakh and other Soviet Republics. As a result, reed biomass has been utilized for pulp and paper products, reed slabs, and fodder yeast in different parts of the Soviet Union. The compact distribution of reed in a particular territory was an indispensable condition to make use of plant feedstocks on a scale necessary, for example, for pulp and paper production. Therefore, an inventory of reed bed areas, and the determination of their distribution and stocks in various regions of the Soviet Union started in 1950 by different research institutes. A comprehensive study of reed beds distribution and its biomass stocks in the Kazakh SSR started in the late 1950s after a string of explorative surveys. Thus, it has been revealed that the Kazakh SSR harbored two-thirds of all reed bed areas of the Soviet Union [24,55]. Sufficiently rich in land and natural resources, the Kazakh SSR became a Republic with the most abundant reed biomass resources within the Soviet Union [79].

In the literature [24,54,55], the provinces accounted for 35–90% of the reed bed area with industrial significance. Owing to the above, reed bed areas in particular provinces were considered to be suitable for mechanized harvesting. This suggested that the area was highly appropriate for industrial utilization and its profitability was undoubtedly high. Although reed biomass had multiple uses, it has not been fully utilized in Kazakhstan for the last half-century. Its most recent harvesting and utilization on an industrial scale was reported at the pulp and paper mill in Kzyl-Orda Province, in the 1970s [55].

## 5.2. Current Development of Reed Resources and Their Utilization

Currently, only certain reed bed areas are partially developed for subsistence use in rural livelihoods, while its biomass harvesting for other commercial purposes, for example, fodder for livestock, construction material for building, and as raw material for pellets production, seldom exists [19,28,31,56,64,80,81]. Next to those applications of harvested reed biomass, reed beds are also used to treat wastewater and oil-contaminated soils from oil industries [82]. Owing to its current marginal utilization, common reed, even today, contributes to settlement development, livestock farming, recreation, and tourism in rural areas of Kazakhstan [28,64,81]. Although having this multipurpose utilization, today, the use of large reed bed areas, however, remains untapped, for example, the large reed bed areas in the Ili Delta are only partly used for grazing and haymaking. At the same time, harvest for other purposes is almost absent [28]. However, grazing of reed beds is connected with management practices, such as burning, which can hardly be considered to be sustainable, and haymaking during summer to stockpile winter fodder, as done in the Ili Delta [64]. These practices can be harmful to reed beds [37,83]. Continuous early and midsummer (April to July) mowing is considered to be a way to combat reed stands, as reported by [39,84,85].

## 5.3. Perspective for Reed Resources Development and Utilization

Next to the provision biomass as fodder for livestock, construction material, among other uses, common reed has been identified as an important renewable resource in the market due to its high biomass productivity [13,19,31,86]. Equally, its ability to grow in water and wet soils contributes

a sustainable way to manage wetlands while, at the same time, providing ecosystem services [28,30,86]. Therefore, in recent years, research and development have explored several new biomass utilizations, for example, for advanced construction materials [87,88], biofuels [89], biogas [90], and most recently bioplastics [91]. Comparative studies, from an economic and ecological point of view, have indicated that material use is often preferable over energetic use of reed biomass [92,93].

So far, innovative reed applications seldom reach industrial scales. Incomplete or ineffective production processes, high harvesting and transportation costs, small underdeveloped markets, non-supportive institutional and political framework conditions, and a limited resource base are among the most critical constraints in Germany, Europe, and elsewhere. Reed stands in Germany, for instance, are small scale, dispersed, and often strictly protected. Thus, even in the well-developed market for thatching, only 15% of the annual demand can be satisfied with domestic sources [94].

In Kazakhstan, contrary vast reed resources are currently hardly used and only in parts under protection. For example, the Ili Delta at Lake Balkhash alone with a recently estimated 211,778 ha of reed beds and 869,097 t of biomass (in submerged reed beds) represents one of the largest contiguous reed stands worldwide [28]. This huge reed biomass potential is presently leaving ample opportunities to expand and improve their sustainable utilization without getting in competition with food and feed production or nature conservation and climate change mitigation goals.

## 6. Discussion and Conclusions

This paper reviewed biomass resources of common reed (*Phragmites australis*) in Kazakhstan with an emphasis on its historical developments, utilization, and prospects. Furthermore, the distribution of *Phragmites* wetlands was explored in Kazakhstan, as they are, by far, the most productive ecosystems there. Therefore, wetlands and reed beds are core areas from which a bioeconomy in such a dryland setting can obtain its feedstock. Although a detailed historical overview of reed resources has been performed, a current reed resources assessment for Kazakhstan is still not available. Therefore, a limitation of the present study is due to very little recent research on this topic and this remains a gap only partly filled by this paper.

Historical reed bed areas distribution in the 13 different provinces of Kazakhstan showed a wide variation ranging from 15,000 ha in Qaraghandy Province to 500,000 ha in Almaty Province (Table 3). This is linked to the presence and conditions of rivers and water bodies that form wetlands in these provinces, since reed beds are mainly concentrated along rivers and their deltas. For example, the Ili River flows through Almaty Province and its delta with one of the country's largest wetlands is also located in this province [28,56,64]. The Ili River and its delta are still rather undisturbed as compared with the Syr Darya Delta, which has experienced degradation in the context of the desiccation of the Aral Sea [62,95]. Accordingly, the relative loss of high yielding reed beds was higher in the Syr Darya Delta in Kyzylorda Province as compared with the loss in the Ili Delta (Table 4). The provinces further north in Kazakhstan, such as Qaragandy, only have small rivers, and therefore smaller reed bed areas. However, in certain provinces, reed beds have been affected by decreasing water levels, and therefore fewer ha of reed beds have been reported.

Similarly, the reed biomass stocks ranged between 75,000 and 5,500,000 t in Qaraghandy and Almaty provinces, respectively (Table 3). Although up-to-date information regarding the distribution of reed beds is still scarce in certain districts, for example, in Atyrau and Mangystau Provinces, within other provinces, the distribution of reed beds and biomass stocks by districts also varied. Possible reasons could be severe conditions for surveying and monitoring the reed bed sites on the most dynamic sites of water bodies such as the coasts of the Caspian Sea, where the water level is gradually decreasing and a broad coastal strip is released.

Several driving forces were outlined for changes in wetlands and reed bed areas in Kazakhstan. Human activities such as grazing and mowing of reed beds, and management practices, such as burning, carried by the local population, have an impact on reed stands. In addition, the variation in the area under reed beds could also be explained by wetland ecosystems recession and their destruction

resulting from activities with agricultural priorities such as the regulation of river runoff for irrigation and drain wetlands to turn them into cultivation and pasture lands or hayfields. Increasing water shortages, due to the expansion of agriculture against the background of rapid population growth and climate change, are considered as the main threats to reed beds in perspective. The later has been predicted to augment water scarcity, thus, negatively interfering, firstly, with the productivity of the reed lands, and then with their sustainability. Additionally, high population growth rate in Central Asia is also leading to an increase in water consumption, resulting in competition for water resources not only between countries but also inside countries between different water consumers such as irrigated agriculture and natural ecosystems.

Past, current, and future utilization of reed resources were inspected for prospect developments of reed resources and their utilization in Kazakhstan. Although reed resources are deemed important, their sustainable use in Kazakhstan is likely limited by the lack of supportive institutional and political framework conditions, and absence of infrastructure capacity for production in scale and small underdeveloped markets. With the recent rise in the number of sites declared as wetlands of international importance, in Kazakhstan, reed as a productive perennial grass that thrives in wetlands could be of economic importance. Its productivity is as high up to 30 t ha$^{-1}$ year$^{-1}$, and its sustainable use could contribute to the overall conservation of reed areas and wetlands. Its prospects in economic terms to the supply of biomass for bioenergy, as well as biobased products, in the future could be of significant importance to rural livelihoods.

The explored potential of reed biomass stock, in Kazakh provinces, is mainly based on studies that were published in the Russian language in the 1950s and 1960s. The results of this literature are presented here for the first time to an international audience. Although there are some variations in the literature concerning the estimates of total areas under reed beds and biomass stocks in individual provinces, the estimates still provide a summary of reed bed distribution at the provincial level from which further research can be conducted.

Overall, Kazakhstan is still likely to host a vast area under reed beds (20% of the total global) [53,54,96–99], as the large area of reed beds (over 211,000 ha) has been previously reported for the Ili Delta alone [25]. Furthermore, large reed bed areas of similar size in the country are the Syr Darya Delta and the Ural Delta, however, the Tenghiz Lakes are relatively small [28,53,54]. Therefore, Kazakhstan is a valuable example to explore the potential of reed beds for bioeconomy. Further research into reed bed utilization and its embedding in landscape planning should, first, concentrate on these deltas. However, as a basis, reed bed area and biomass data are needed, and this is the gap filled by this publication. However, in this review the presented consolidated data on reed bed areas with ramifying numbers in some provinces limit comprehensive statements. The information is mostly indicative, and therefore requires clarification. Therefore, the use of remote sensing and GIS (geographic information system) technologies along with ground truthing could enhance and renew the estimation of reed biomass in Kazakhstan in, as it was done for the delta of Ili River during 2014 and 2015 [28,100]. Similar to an updated country-wide assessment of reed resources using remote sensing methods is recommendable. Moreover, the potential benefits of reed for sustainable utilization still remain to be fully realized.

The biomass productivities of the reed beds listed, here, for Kazakhstan are in the same range as fast-growing tree plantations. New plantations of fast-growing trees would require using land that often could also be used as farmland for crops. The reed beds do exist, and therefore biomass harvest from those reed beds does not occupy cropland.

**Author Contributions:** Conceptualization, A.B. and N.T.; formal analysis, A.B. and N.T.; literature search, A.B. and S.N.; draft preparation and editing, A.B., N.T., V.B., and R.W.; visualization, A.B. and N.T.; map design, R.S.; supervision, R.W. and S.N. All authors have read and agreed to the published version of the manuscript.

**Funding:** This study received funding support from the German Academic Exchange Service (DAAD).

**Acknowledgments:** We are thankful for the technical support of team members of National Library of Republic of Kazakhstan in Almaty for their support during the literature search.

**Conflicts of Interest:** The authors declare no conflict of interest.

## Appendix A. Reed Distribution and Its Biomass Stocks by Administrative Levels in the 1960s

*Appendix A.1. Almaty Province*

The Alma-Ata and Taldykorgan Provinces were unified and renamed as Almaty Province in 1997. This province contains the largest reed bed area as compared with other provinces (see Table 3). Here, the major reed bed area is the region of the Ili River Delta, the western and eastern shores of Lake Balkhash located in the Balkhash District. Their reed bed areas are sumarized in Table A1. Other important reed bed areas in Almaty Province are the coasts of the Lakes Alakol and Sasykkol, and the middle reaches of the Ili River.

**Table A1.** Reed bed areas in Balkhash District (source [54], 1964).

| Natural Landmark Names | The Total Reed Bed Area (ha) | Area of Production Reed Bed (ha) |
|---|---|---|
| Akkol Lake | 33,100 | 9900 |
| Aksiyyr | 30,000 | 15,000 |
| Topar | 9000 | 2700 |
| Kokkul | 36,100 | 7200 |
| Ak-Soya | 21,000 | 6800 |
| Zhuzbay | 20,000 | 20,000 |
| Zheltorangy | 11,300 | 4500 |
| Zhideli (right bank) | 26,000 | 7800 |
| Beginning of the duct Zhideli | 12,600 | 1200 |
| Bokkore (left bank) Zhideli | 16,500 | 3300 |
| Araltobe | 2000 | 800.0 |
| Kur-Ili | 69,700 | 13,900 |
| Bosingen | 6700 | 1500 |
| Arkhar on the Zhideli River | 8400 | 4200 |
| Kokkol | 11,500 | 3400 |
| Muzdybai Arystan | 20,300 | 12,200 |
| Tuzdykol and islands along the Ili River | 500.0 | 200.0 |
| Along the banks of the Ili River and the banks of Lake Balkhash | 28,300 | 28,300 |
| Total | 363,200 | 142,700 |

The estimated total area of reed beds was approximately 500,000 ha, including 457,600 ha recorded by the Land Management Department of Alma-Ata Regional Agricultural Administration [54]. Up to 90%, i.e., 450,000 ha out of the total array of reed beds were accounted as available for industrial purposes with a total stock of 4.9 million t. However, additional surveys following government decrees and contracts between stakeholders in an association "Agrolesoproekt" examined the reed beds of the Alma-Ata Province and resulted in 639,000 ha (Table A2).

**Table A2.** Reed bed areas and reed stocks in districts of former Alma-Ata Province (source [54], 1964).

| District Names | Area (ha) | Biomass Stocks (t) |
|---|---|---|
| Aksu | 1400 | 400 |
| Alakol | 68,400 | 391,300 |
| Balkhash | 433,100 | 1,340,600 |
| Ili | 37,400 | 17,700 |
| Karatal | 39,000 | 108,300 |
| Panfilov | 16,600 | 64,400 |
| Uigur | 16,200 | 33,000 |
| Shelek | 12,400 | 26,900 |
| Enbekshi-Kazakh | 14,500 | 40,300 |
| Total | 639,000 | 2,022,900 |

The overall surveyed reed bed area of Almaty Province was further divided into three groups based on the principle of possibility and profitability of mechanized reed harvesting (Table A3) [54]. Of the total area, 35.7% was accounted as suitable for industrial use with the reed bed of the first and second groups, which were available for mechanized harvesting with a minimum yield of 4 t per ha. Meanwhile, the third group of the remaining area, which was considered unsuitable for industrial use, included 125 ha of monospecies cattail beds (with a total stock of 589 t) and 1116 ha of the mixed reed-cattail beds (with a total cattail stock of 2072 t).

**Table A3.** Reed beds distribution by districts and by groups available for harvesting in Almaty Province (source [54], 1964).

| District Names | Groups for Production Utilization | | | | | | | |
| --- | --- | --- | --- | --- | --- | --- | --- | --- |
| | Industrial | | | | | | Agricultural | |
| | First | | Second | | Total of Two Groups | | Area (ha) | Biomass Stock (t) |
| | Area (ha) | Biomass Stock (t) | Area (ha) | Biomass Stock (t) | Area (ha) | Biomass Stock (t) | | |
| Aksui | - | - | - | - | - | - | 1375 | 412 |
| Alakol | 1020 | 4284 | 53,820 | 357,582 | 54,840 | 361,866 | 13,570 | 29,459 |
| Balkhash | - | - | 142,875 | 958,728 | 142,875 | 958,728 | 290,235 | 381,850 |
| Ili | 100 | 400 | 1281.6 | 10,265 | 1381.6 | 10,665 | 2362 | 7050 |
| Karatal | - | - | 6995 | 43,369 | 6995 | 43,369 | 31,960 | 64,895 |
| Panfilov | 1010 | 10,165 | 3507 | 25,993 | 4517 | 36,158 | 12,310 | 28,266 |
| Uigur | - | - | 1871 | 9725 | 1871 | 9725 | 14,310 | 24,210 |
| Shelek | 994.6 | 4095 | 1096.8 | 8061 | 2091.4 | 12,756 | 10,367.7 | 14,178 |
| Enbekshi-Kazakh | 199 | 1393 | 1551.4 | 10,822 | 1750.4 | 12,215 | 12,787.3 | 208,098 |
| Total | 3323.6 | 20,937 | 212,997.8 | 1,424,545 | 216,321.4 | 1,445,482 | 38,900.7 | 378,448 |

*Appendix A.2. Kyzylorda Province*

The Kzyl-Orda Province was renamed as Kyzylorda Province in 1997. The Syr Darya River is the lifeline of that province. The delta and the northern part of Aral Sea also belong to this province. Reed beds mainly occur along the river and its delta. The Syr Darya River periodically floods almost all reed beds in this province during its spring floods. According to [26], as well as to the generalized data (see Table 3) given by [54], the total area occupied by reed beds was 400,000 ha with a total yield of up to 5 million t.

Kzyl-Orda Province was surveyed by the central integrated expedition of the Ministry of Agriculture and Procurement of the Kazakh SSR, in 1952. Areas that included reed were described for 89,300 ha. Additionally, surveys on the feedstock for the pulp and paper mill in Kzyl-Orda City were carried out in 1957 and resulted in an area of 35,300 ha of reed beds across Kzyl-Orda Province (Table A4), which is much less than in the preceding study.

**Table A4.** Reed stocks and distribution in districts of the Kzyl-Orda Province, in 1957, modified after [54].

| District Names | Areas (ha) | | | Biomass Stocks (t) |
| --- | --- | --- | --- | --- |
| | Reed Beds | Cattail Beds | Total | |
| Karmakshi and Dzhalagash | 5000 | - | 5000 | 60,000 |
| Dzhalagash and Teren-Ozek | 19,900 | 2800 | 22,700 | 238,800 |
| Syrdarya | 7100 | 500 | 7600 | 85,200 |
| Total | 32,000 | 3300 | 35,300 | 384,000 |

In the late 1950s, the research topic "Biological features of reed in connection with its economic use" was proposed for execution by the State Planning Committee of the Kazakh SSR. The Institute of Botany took the lead and reported a total area under reed and cattail beds of up to 218,270 ha with the feedstock reserves of up to 1.7 million t (Table A5) [101]. Those numbers were close to the results from 1952 (see above).

**Table A5.** Reed stocks and distribution in natural landmarks of Kzyl-Orda Province 1959–1960 (source [54]).

| Natural Landmark Names | Total Area of the Array (ha) | Area of Reed Beds for Industrial Use (ha) | Gross Stock of the Feedstock (t) | Production Stock (t) |
|---|---|---|---|---|
| Bakaly-Kopa | 61,140 | - | 267,793.0 | - |
| Kara-Ozek | 40,330 | 27,608 | 458,020.0 | 321,780.0 |
| Koksu-Kerkelmes | 36,130 | 20,012 | 33,5461.0 | 236,252.0 |
| Shieli-Baygakum | 24,300 | 9002 | 149,042.0 | 108,468.5 |
| Kara-Ketken | 19,100 | 11,554 | 163,570.0 | 115,018.0 |
| Tartugai | 11,380 | 2344 | 41,428.0 | 30,236.5 |
| Zhanakorgan | 9110 | 4032 | 59,082.0 | 44,687.0 |
| Dalakol | 8810 | 7048 | 117,177.4 | 30,335.0 |
| Dzhalagash | 5330 | 3738 | 63,306.0 | 44,173.0 |
| Birkazan | 2640 | 2100 | 38,666.0 | 29,000.0 |
| Total | 218,270 | 87,438 | 1,693,545.4 | 1,010,550.0 |

In 1959, the Institute of Economics reported its data on reed bed areas and stocks in districts of Kzyl-Orda Province (Table A6). The estimated total area of reed beds with industrial importance was 126,000 ha with a total biomass stock of up to 1.9 million t.

**Table A6.** Reed areas of industrial significance and their stocks in districts of Kzyl-Orda Province in 1959 (source [54]).

| District Names | Area (ha) | Biomass Stocks (t) |
|---|---|---|
| Aral | 25,000 | 375,000 |
| Dzhalagash | 15,000 | 225,000 |
| Kazaly | 25,000 | 375,000 |
| Karmakshy | 10,000 | 150,000 |
| Teren-Ozek | 15,000 | 225,000 |
| Syrdarya | 16,000 | 240,000 |
| Shieli | 15,000 | 225,000 |
| Zhanakorgan | 5000 | 75,000 |
| Total | 126,000 | 1,890,000 |

*Appendix A.3. Atyrau Province*

The former Guryev Province was renamed as Atyrau Province in 1991. Reed beds mainly occur on the coast of the Caspian Sea, as a broad coastal strip has been exposed due to a gradual decrease in water level. There, at least 400,000 ha of reed beds were reported. The areas of 275,000 ha under reed as found by [54,98] are not exaggerated.

Furthermore, reed beds were also recorded in the lower reaches of the Emba and Sagiz Rivers and along the Ural River upstream from Atyrau City, as well as in its delta. A sea bulrush heavily litters reed beds of Atyrau Province, and cattail is partly mixed into the *Phragmites* stands. These common reed and cattail mixed stands are mainly confined to the northern coast of the Caspian Sea.

There is no latest information about reed bed sites in districts of the province. According to the materials of the Institute of Economics of the Academy of Sciences of the Kazakh SSR, there are 180,000 ha of areas occupied by reed bed of industrial significance (Table A7).

**Table A7.** Reed areas of industrial significance and their stocks in districts of Guryev Province in 1959 (source [54]).

| District Names | Area (ha) | Biomass Stocks (t) |
|---|---|---|
| Baksai | 25,000 | 375,000 |
| Guryev | 15,000 | 225,000 |
| Dengiz | 40,000 | 600,000 |
| Zhilakosyn | 20,000 | 300,000 |
| Kyzyl-Koga | 20,000 | 300,000 |
| Novobogatin | 30,000 | 450,000 |
| Makat | 15,000 | 225,000 |
| Shevchenko | 15,000 | 75,000 |
| Total | 180,000 | 2,550,000 |

*Appendix A.4. Dzhambul Province*

The main reed beds in this province occur along the banks of the Shu River in Kokterek District in the north of the province and 97,000 ha of reed bed areas with a total stock of 1.5 million t were reported by the Institute of Economics of the Academy of Sciences of the Kazakh SSR. Reed beds appropriate for industrial utilization accounted for 80% of that total area. Approximately 120,000 ha of reed bed areas with an annual yield of 1.2 million t were reported by the Interim Scientific and Technical Commission of the State Scientific and Technical Committee under the Council of Ministers of the USSR (1961) [102]. A third source, the Dzhambul Regional Agricultural Administration reported 88,800 ha of reed beds with an average yield of 12 t ha$^{-1}$ year$^{-1}$ (Table A8) [54]. Only this source divided the reed bed areas per district.

**Table A8.** Reed bed areas and reed stocks in districts of Dzhambul Province (source [54], 1964).

| District Names | Area (ha) | Biomass Stocks (t) |
|---|---|---|
| Dzhambul | 3100 | 37,400 |
| Kokterek | 61,700 | 740,000 |
| Kordai | 3400 | 41,400 |
| Lugovoi | 2100 | 26,200 |
| Merke | 2000 | 25,000 |
| Sarysu | 2000 | 24,600 |
| Sverdlov | 2700 | 33,400 |
| Talas | 7000 | 84,500 |
| Shu | 3800 | 46,500 |
| Total | 88,800 | 1,059,000 |

*Appendix A.5. Turkistan Province*

The former Chimkent Province, firstly, was renamed back to its old name South-Kazakhstan Province in 1992. In 2018, it was renamed as Turkistan Province. There are reed beds across an area of 100,000 ha [54]. Reed beds are mainly concentrated in the Turkestan, Shauldir, Sozak, and Arys Districts, which are located along the Syr Darya River. Reed bed areas of 87,500 ha with an average yield of 8 t ha$^{-1}$ year$^{-1}$ were recorded by the Land Management Department of Shymkent Regional Agricultural Administration and the Institute of Botany in 1959 (Table A9).

*Appendix A.6. Qostanai Province*

The reed beds in this province are located along the Turgai River and around the Lakes Sarymoni, Zharkul, Sarykopa, and Kamyshlykul. The estimated total area of reed beds was approximately 85,000 ha [54]. The Provisional Scientific and Technical Commission of the State Scientific and Technical Committee under the Council of Ministers of the USSR estimated an area 80,000 ha of reed bed with an annual yield of 480,000 t for this province [103].

**Table A9.** Reed bed areas and reed stocks in districts of former Chimkent Province in 1959 (source [54]).

| District Names | Area (ha) | Biomass Stocks (t) |
|---|---|---|
| Arys | 10,500 | 84,000 |
| Keles | 2600 | 21,000 |
| Frunze | 8100 | 65,000 |
| Turkestan | 13,100 | 105,000 |
| Shauldir | 23,800 | 190,000 |
| Sozak | 29,300 | 235,000 |
| Total | 87,500 | 700,000 |

Meanwhile, the Institute of Economics of the Academy of Sciences of the Kazakh SSR determined the area of 58,000 ha (in the five districts of the province) of reed beds with a total biomass stock of 695,000 t [53]. However, the Land Management Department of Kostanai Regional Agricultural Administration recorded 80,800 ha of reed bed areas with an average yield of 6 t ha$^{-1}$ year$^{-1}$ for the whole province (Table A10).

**Table A10.** Reed bed areas and reed stocks in districts of Qostanai Province in 1959 (source [54]).

| District Names | Area (ha) | Biomass Stocks (t) |
|---|---|---|
| Amangeldi | 6500 | 39,000 |
| Dzhangeldi | 14,200 | 86,000 |
| Dzhetygara | 90 | - |
| Zatobol | 2700 | 16,400 |
| Kamyshin | 2100 | 12,500 |
| Karabalyk | 6300 | 38,500 |
| Karasu | 13,100 | 79,800 |
| Kostanai | 600 | 3600 |
| Mendykara | 8100 | 49,000 |
| Oktyabr | 500 | 3000 |
| Ordzhonikidze | 40 | - |
| Presnogorkov | 2500 | 10,000 |
| Semiozerni | 1200 | 7400 |
| Taranov | 400 | 1200 |
| Ubagan | 5500 | 33,000 |
| Urit | 2800 | 17,000 |
| Uzynkol | 5500 | 33,000 |
| Fedorov | 8400 | 50,400 |
| Total | 80,800 | 480,000 |

*Appendix A.7. Aktyubinsk Province*

The reed beds are mostly concentrated along the Rivers Temir, Emba, and Big Hobda. The total area was 70,000 ha of reed bed with an annual yield of 560,000 t mainly in the Uil, Shalkar, Dzhurun and Karabutak Districts [102]. However, the Land Management Department of Aktobe Regional Agricultural Administration recorded 48,700 ha of reed bed areas with an average yield of 6 t ha$^{-1}$ year$^{-1}$ (Table A11). Reed beds with industrial significance accounted for 40–60% of the total reed bed area [54]. The rest of reed beds that were not listed in Table A8 occupied 12,000 ha in the remaining districts of Aktubinsk Province.

**Table A11.** Reed bed areas and reed stocks in districts of Aktubinsk Province in 1959 (source [54]).

| District Names | Area (ha) | Biomass Stocks (t) |
|---|---|---|
| Dzhurun | 50 | 300 |
| Karabutak | 1300 | 7800 |
| Irgiz | 47,000 | 282,000 |
| Novorossiysk | 100 | 900 |
| Uil | 200 | 1000 |
| Total | 48,650 | 292,000 |

*Akmola Province*

The former Tselinograd Province (**A**) was renamed as Akmola Province in 1992. Furthermore, three southern districts of the current North Kazakhstan Province (Zerendi, Schuchin, and Enbekshilder, which until 1997 were part of the abolished Kokchetav Province (**B**)) were consigned to the Akmola Province in 1999.

**A**. Reed beds in Tselinograd Province are concentrated in the system of the Lakes Tenghiz-Korgalzhyn, Vishnevskoe, Erkinshilikskoye, Kiylinskoye, and Novocherkasskoye. This lake system comprised at least 25,000 ha of reed beds with a total yield of 180,000 t year$^{-1}$, which is half of the reed bed area of the whole province [54]. According to [26], the total area under reed beds was 50,000 ha, while the Land Management Department of Tselinograd Regional Agricultural Administration recorded 37,600 ha with an average yield of 8 t ha$^{-1}$ year$^{-1}$ (Table A12).

**B**. Despite the presence of mountains in Kokchetav Province, there are many lakes, along which banks reed beds are distributed across an area of 55,000 ha with a total yield of 380,000 t year$^{-1}$ [54]. Almost all reed bed arrays located in Ayrtau, Kokshetau, and Enbekshilder Districts were stands available for industrial use, making up 80%. According to [26], the total area under reed beds was 60,000 ha, while the Land Management Department of Kokshetav Regional Agricultural Administration recorded 56,000 ha with an average yield of 6 t ha$^{-1}$ year$^{-1}$.

**Table A12.** Reed bed areas and reed stocks in districts of former Tselinograd Province in 1959 (source [54]).

| District Names | Area (ha) | Biomass Stocks (t) |
|---|---|---|
| Tselinograd | 3800 | 30,700 |
| Atbasar | 500 | 4400 |
| Barankol | 5000 | 40,000 |
| Vishnev | 1900 | 15,200 |
| Kalinin | 200 | 1000 |
| Korgalzhyn | 17,800 | 145,400 |
| Novocherkas | 6400 | 51,200 |
| Shortandy | 800 | 6400 |
| Erkinshilik | 1200 | 13,000 |
| Total | 37,600 | 307,300 |

*Appendix A.8. East Kazakhstan Province*

In 1997, the former Semipalatinsk Province (**C**) was merged into East Kazakhstan Province (**D**).

**C**. Major reed beds in the former Semipalatinsk Province are concentrated along the coast of Alakol, Sasykkol, and Uyaly Lakes, which are the largest lakes of that former province. Up to 100,000 ha of reed bed areas with a total yield of approximately 1 million t year$^{-1}$ was reported by the Provisional Scientific and Technical Commission of the State Scientific and Technical Committee under the Council of Ministers of the USSR [102]. However, the Land Management Department of Semipalatinsk Regional Agricultural Administration recorded 46,400 ha with an average yield of 8 t ha$^{-1}$ year$^{-1}$ (Table A13) [54].

Meanwhile, 60,594 ha of reed bed areas with a total reed stock of 171,000 t were recorded based on surveys of the "Agrolesoproekt" association in 1961. The first and second Groups of reed beds make up 21,900 ha, i.e., 36.2% of the total area with 121,200 t of total stock [26].

**D**. Reed beds are mainly concentrated on the coast of Lake Zaysan and in the lower reaches of the Black Irtysh River. The area was approximately 40,000 ha, while the Land Management Department of Regional Agricultural Administration recorded 32,218 ha with an average yield of 10 t ha$^{-1}$ year$^{-1}$ (Table A14) [54]. The Institute of Economics of the Academy of Sciences of the Kazakh SSR determined an area of 22,000 ha (in four districts of the province) of reed bed available for industrial use with a total reed stock of 220,000 t [53].

**Table A13.** Reed bed areas and reed stocks in districts of former Semipalatinsk Province in 1959 (source [54]).

| District Names | Area (ha) | Biomass Stocks (t) |
| --- | --- | --- |
| Ayaguz | 12,000 | 96,000 |
| Borodulikhin | 150 | 1000 |
| Zharmin | 150 | 1200 |
| Kokpekty | 800 | 6400 |
| Novopokrov | 700 | 5600 |
| Urdzhar | 30,500 | 224,000 |
| Makanchi | 2000 | 16,000 |
| Char | 100 | 800 |
| Total | 46,400 | 351,000 |

**Table A14.** Reed bed areas and reed stocks in districts of former East Kazakhstan Province in 1959 (source [54]).

| District Names | Area (ha) | Biomass Stocks (t) |
| --- | --- | --- |
| Bolshenarym | 2300 | 22,800 |
| Zaysan | 9300 | 93,100 |
| Zyryanov | 1700 | 17,100 |
| Kurchum | 11,100 | 111,300 |
| Markakol | 2600 | 26,500 |
| Samar | 4200 | 41,200 |
| Total | 31,200 | 312,000 |

*Appendix A.9. North Kazakhstan Province*

This province today consists of the former North Kazakhstan Province and the former Kokchetav Province, which was merged into the North Kazakhstan Province in 1997.

Major reed beds occurred along the Ishim River and on the coast of lakes in the province. Approximately 20,000 ha of reed bed areas with a total yield of 120,000 t year$^{-1}$ were reported by the Interim Scientific and Technical Commission of the State Scientific and Technical Committee under the Council of Ministers of the USSR (1961) [102]. However, in [26], it was claimed that areas reach up to 35,000 ha, while the Land Management Department of Regional Agricultural Administration recorded 29,600 ha of reed bed areas with an average yield of 5 t ha$^{-1}$ year$^{-1}$ (Table A15) [54].

Meanwhile, the materials of the Institute of Economics of the Academy of Sciences of the Kazakh SSR contain only a brief indication of the presence of 5000 ha of reed beds available for industrial use [53]. This number is underestimated.

**Table A15.** Reed bed areas and reed stocks in districts of former North Kazakhstan Province in 1959 (source [54]).

| District Names | Area (ha) | Biomass Stocks (t) |
| --- | --- | --- |
| Konyukhov | 2100 | 10,800 |
| Lenin | 1100 | 6000 |
| Mamlut | 9900 | 49,500 |
| Oktyabr | 2000 | 9800 |
| Poludin | 600 | 3000 |
| Presnov | 6300 | 31,400 |
| Sovet | 7500 | 37,900 |
| Total | 29,500 | 148,400 |

*Appendix A.10. West Kazakhstan Province*

The former Uralsk Province was renamed as West Kazakhstan Province in 1992. Approximately 15,000 ha of reed bed areas with an average yield of 7 t ha$^{-1}$ year$^{-1}$ were recorded by the Land Management Department of Regional Agricultural Administration (Table A16).

**Table A16.** Reed bed areas and reed stocks in districts of former Uralsk Province in 1959 (source [54]).

| District Names | Area (ha) | Biomass Stocks (t) |
|---|---|---|
| Dzhambeity | 1400 | 11,200 |
| Dzhangalin | 2000 | 16,000 |
| Kamen | 2000 | 16,000 |
| Furmanov | 5000 | 40,000 |
| Chapaev | 3000 | 24,000 |
| Total | 13,400 | 107,200 |

However, the Institute of Economics of the Academy of Sciences of the Kazakh SSR reported 40,000 ha of reed beds with a total yield of 400,000 t year$^{-1}$ [53]. The low area of 15,000 ha, given in [54], in contrast to 40,000 ha is due to large areas that were converted in hayfields and cropland during 1959–1964.

*Appendix A.11. Pavlodar Province*

Reed beds, in Pavlodar province, were determined along the Irtysh River in an area of 40,000 ha with a total yield of 240,000 t year$^{-1}$. However, 15,000 ha of the total reed bed areas were recorded by [26]. In comparison, 11,900 ha of reed beds with an average yield of 5 t ha$^{-1}$ year$^{-1}$ were reported by the Land Management Department of Pavlodar Regional Agricultural Administration (Table A17) [54]. Reed beds of industrial significance accounted for approximately 80%.

**Table A17.** Reed bed areas and reed stocks in districts of Pavlodar Province in 1959 (source [54]).

| District Names | Area (ha) | Biomass Stocks (t) |
|---|---|---|
| Bayanaul | 200 | 800 |
| Beskaragai | 50 | 300 |
| Ermakov | 350 | 1700 |
| Irtysh | 1400 | 7000 |
| Kuibyshev | 3200 | 16,200 |
| Lebyazhen | 200 | 1200 |
| Lozov | 30 | - |
| Maxim Gorki | 4600 | 23,000 |
| Pavlodar | 10 | - |
| Urlutin | 1400 | 7400 |
| Tsuryupin | 400 | 2400 |
| Total | 11,840 | 60,000 |

*Appendix A.12. Qaraghandy Province*

Major reed beds are relatively small in this province and are mainly concentrated along the Taldy River. Reed bed areas were unevenly distributed in individual districts. Only 10,150 ha were reported by the Interim Scientific and Technical Commission of the State Scientific and Technical Committee under the Council of Ministers of the USSR (1961) [102]. However, the Land Management Department of Karaganda Regional Agricultural Administration recorded 14,430 ha of reed bed areas with an average yield of 5 t ha$^{-1}$ year$^{-1}$ (Table A18) [54].

**Table A18.** Reed bed areas and reed stocks in districts of Qaraghandy Province in 1959 (source [54]).

| District Names | Area (ha) | Biomass Stocks (t) |
|---|---|---|
| Voroshilov | 500 | 2500 |
| Zhana-Arka | 640 | 3200 |
| Karkaralyn | 200 | 1000 |
| Kounrad | 3200 | 16,000 |
| Kuv | 540 | 2700 |
| Nurin | 4160 | 20,500 |
| Osokarov | 370 | 1900 |
| Telman | 900 | 4500 |
| Ulutau | 3900 | 14,700 |
| Total | 14,410 | 67,000 |

Moreover, 3600 ha of reed bed areas belonging to the third group with a total reed stock of 5900 t were recorded in Kounrad District based on surveys of the "Agrolesoproekt" Association.

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
