# Peer review of "Biomass Resources of Phragmites australis in Kazakhstan: Historical Developments, Utilization, and Prospects"

_resources, doi:10.3390/resources9060074_

Round 1

Reviewer 1 Report

Table 1: Biomass is expressed in t/ha or kg/m2 or g/m2. The unit containing time factor, year in this case, is productivity. Biomass represents a dry weight of biomass at a given time. I recommend to re-evaluate the results for biomass. these numbers are too low, especially for Danube Delta. I would expect values of about 40-60 t/ha (it is 4000 - 6000 g/m2).

Figure 2: most numbers are included in Table 1 so this figure can be omitted

Tables 2 and 3: How much are these data valid now? The numbers are more than 50 years old. Are there any estimates for current situation?

The manuscript provides  very good results from the past but the newest (and only few) results are from 2012. It is difficult to speak about perscpectives when no information about the current situation is given.

Before publication, the authors need to add information about the current situation. For example Table 22 should also include numbers from about 2018-2019. Otherwise it is a manuscript abut the history and not perspectives.

Author Response

Response to Reviewer 1 Comments

Authors are grateful to the Reviewer for helpful feedback that helped to improve the quality of the manuscript. Authors have carefully responded to all points and have revised the manuscript.

Point 1: Table 1: Biomass is expressed in t/ha or kg/m2 or g/m2. The unit containing time factor, year in this case, is productivity. Biomass represents a dry weight of biomass at a given time. I recommend to re-evaluate the results for biomass. these numbers are too low, especially for Danube Delta. I would expect values of about 40-60 t/ha (it is 4000 - 6000 g/m2).

Response 1: Authors thankful for this comment, which has been positively perceived. Authors have corrected the wordings according to the Reviewer’s notes.

In Lines 187-188 (136-137): Biomass in Table 1 is expressed in t/ha and the unit containing year as a time factor is productivity given as “Harvested biomass” which was inherited from the source (Köbbing et al., 2013). Thereby, to make refinements, the terms “Harvested biomass” and "Total yield" have been renamed to "Average productivity" and "Total biomass" correspondingly.  The results for biomass were revised and updated values are given. According to Brix et al., 2001. above-ground biomass value for Danube Delta (Romania) evaluated as 26 t/ha (or it is 2590 g/m2) and biomass of rhizome 41 t/ha (to it is 4090 g/m2) while in sum reach 6680 g/m2.

Point 2: Figure 2: most numbers are included in Table 1 so this figure can be omitted

Response 2: Figure 2 is omitted.

Point 3: Tables 2 and 3: How much are these data valid now? The numbers are more than 50 years old. Are there any estimates for current situation?

Response 3: Authors agree that the numbers in Table 2 and 3 are more than 50 years old. However, it is within the best attempt to present the most recent available data on reed biomass resources in Kazakhstan. For several decades now reed beds in most parts of Kazakhstan have been neglected in research. In dryland regions, wetlands and reed bed areas are by far the most productive ecosystems. Therefore, wetlands and reed beds are core areas from which a bio-economy in such a dryland setting can obtain its feedstock. This highlights the importance of reed beds, in particular in dryland countries. In addition, Kazakhstan has so many reed beds (20% of the total global). Therefore, Kazakhstan is a valuable example to explore the potential of reed beds for bio-economy. As a basis, reed bed area and biomass data are needed. But such data do not exist - and this is the gap filled by this paper.

Point 4: The manuscript provides very good results from the past but the newest (and only few) results are from 2012. It is difficult to speak about perspectives when no information about the current situation is given.

Response 4: Authors appreciate this positive assessment of the manuscript. Authors have to admit that there is no current reed resources assessment for Kazakhstan is available. Although very little has been published in the past authors communicate the most recent results with emphasis on a historical overview of reed resources in the largest wetland complex in the region: the Ili-Delta.

Point 5: Before publication, the authors need to add information about the current situation. For example, Table 22 should also include numbers from about 2018-2019. Otherwise it is a manuscript about the history and not perspectives.

Response 5: In-Lines 355-388 (212-244): to meet the recommendations and improve the manuscript authors developed a new Subsection 3.3. A showcase of a historical overview of reed resources in the Ili Delta. Authors have added available information about the current situation for the Ili Delta. The Ili Delta is the only reed bed area where the size and distribution of wetlands and reed bed areas were studied several times between the 1960s and today. No more recent data are available for other red bed areas.

Please see also in the attachment.

Reviewer 2 Report

Manuscript concerns distribution of reed resource  potential in wetland areas of Kazakhstan and the prospects for its sustainable  utilization. But in Reviewer opinion this manuscript does not fit to the aim and scope of the Journal.

Paper is very interesting and well prepared that’s why I suggest Authors to submit this manuscript to more appropriate Journal.

I do not recommend this work to publication in Resources Journal.

Author Response

Response to Reviewer 2 Comments

Authors are grateful to the Reviewer for the positive assessment of our work and feedback.

Point 1: Manuscript concerns distribution of reed resource potential in wetland areas of Kazakhstan and the prospects for its sustainable utilization. But in Reviewer opinion this manuscript does not fit to the aim and scope of the Journal.

Response 1: Authors thankful to the Reviewer for the careful review of the manuscript. Authors believe that the topic is quite specific and paper is fit to the aim and scope of the Journal with emphasis on Special Issue. Moreover, the authors stress that worldwide there is a growing interest in Reed resources.

In dryland regions, wetlands and reed bed areas are by far the most productive ecosystems. Therefore, wetlands and reed beds are core areas from which a bio-economy in such a dryland setting can obtain its feedstock. This highlights the importance of reed beds, in particular in dryland countries. In addition, Kazakhstan has so many reed beds (20% of the total global). Therefore, Kazakhstan is a valuable example to explore the potential of reed beds for bio-economy. As a basis, reed bed area and biomass data are needed. But such data do not exist - and this is the gap filled by this paper.

Point 2: Paper is very interesting and well prepared that’s why I suggest Authors to submit this manuscript to more appropriate Journal.

Response 2: Authors appreciate the positive assessment of the paper and suggestion. However, they believe that paper is fully fit for the special issue of the Resources.

Point 3: I do not recommend this work to publication in Resources Journal.

Response 3: Authors cannot accept this comment and rebut it.

Please see also in the attachment

Reviewer 3 Report

Baibagyssov and co-authors present an analysis of the distribution of reed resource potential in wetland areas of 13 provinces of Kazakhstan. The outcomes are interesting and might contribute to advancing our knowledge in the use of common reed as a valuable source of renewable biomass.

However, my main criticism concerns the following points:

- as it stands, the paper reads like a regional technical report with a lot of local information. I think there needs to be a partial rewrite of the introduction, discussion, and conclusions to convert the manuscript into a more interesting scientific paper.

-  the analysis is based on outdated data. To evaluate the actual potential use of reed, an updated picture of its distribution and biomass availability is necessary.

- A deeper and speculative interpretation of the presented data is needed, in a way to better evaluate their spendibility for practical application in the use of reed biomass

Some specific comments:

  • Lines 89-171: this whole paragraph is partially out of topic, please consider synthesize this part and maintain only the information necessary in the context of the present review.
  • Line 194: in some areas around the world (e.g. North America and Oceania), Phragmites is an invasive species
  • Table 1: explain the difference between the two types of data reported here, i.e. harvested biomass and total yield. Is total yield the product of reed bed area per harvested biomass?
  • Tables 1 and 2: what is the difference between the term “harvested biomass” (used in Table 1) and “biomass productivity” (used in Table 2)? The unit of measure is the same (i.e. t ha-1 yr-1).
  • From line 220: section 4 contains a lot of detailed information about the reed beds areas distribution in different provinces of Kazakh SSR. I think this whole paragraph may be scarcely appealing for an international audience
  • Table 22: the authors compare here the reed area and biomass between the ‘60s and 1990/1993. No more recent data available? What is the present situation?

Author Response

Response to Reviewer 3 Comments

Authors are grateful to the Reviewer for helpful feedback that helped to improve the quality of the manuscript. Authors have carefully responded to all points and have revised the manuscript accordingly.

Point 1: Baibagyssov and co-authors present an analysis of the distribution of reed resource potential in wetland areas of 13 provinces of Kazakhstan. The outcomes are interesting and might contribute to advancing our knowledge in the use of common reed as a valuable source of renewable biomass.

Response 1: Authors thankful to the Reviewer for this positive assessment of the manuscript.

Point 2: However, my main criticism concerns the following points:

- as it stands, the paper reads like a regional technical report with a lot of local information. I think there needs to be a partial rewrite of the introduction, discussion, and conclusions to convert the manuscript into a more interesting scientific paper.

Response 2: Authors thankful for this comment. During the major revision, authors have partially rewrite the introduction, discussion, and conclusions to convert the manuscript into a more interesting scientific paper. Moreover, the structure of the manuscript has been carefully revised and modified. Sections with a lot of local information have been removed.

Point 3: -  the analysis is based on outdated data. To evaluate the actual potential use of reed, an updated picture of its distribution and biomass availability is necessary.

Response 3: Authors agree that the analysis is based on outdated data. However, it is within the best attempt to present the most recent available data on reed biomass resources in Kazakhstan. For several decades now reed beds in most parts of Kazakhstan have been neglected in research. To evaluate the actual potential use of reed an updated picture is provided for the Ili-Delta. In-Lines 355-388 (212-244): to meet the recommendations and improve the manuscript authors developed a new Subsection 3.3. A showcase of a historical overview of reed resources in the Ili Delta. The Ili Delta is the only reed bed area where the size and distribution of wetlands and reed bed areas were studied several times between the 1960s and today.

Point 4: The manuscript provides very good results from the past but the newest (and only few) results are from 2012. It is difficult to speak about perspectives when no information about the current situation is given.

Response 4: Authors appreciate this positive assessment of the manuscript. Authors have to admit that there is no current reed resources assessment for Kazakhstan is available. Although very little has been published in the past authors communicate the most recent results with emphasis on a historical overview of reed resources in the largest wetland complex in the region: the Ili-Delta.

In dryland regions, wetlands and reed bed areas are by far the most productive ecosystems. Therefore, wetlands and reed beds are core areas from which a bio-economy in such a dryland setting can obtain its feedstock. This highlights the importance of reed beds, in particular in dryland countries. In addition, Kazakhstan has so many reed beds (20% of the total global). Therefore, Kazakhstan is a valuable example to explore the potential of reed beds for bio-economy. As a basis, reed bed area and biomass data are needed. But such data do not exist - and this is the gap filled by this paper.

Point 5: - A deeper and speculative interpretation of the presented data is needed, in a way to better evaluate their spendibility for practical application in the use of reed biomass

Response 5: During the major revision, authors have been carefully revised and modified the structure of the manuscript. Within Section 5. Past, current and prospect developments of reed resources and their utilization in Kazakhstan in Lines 646-719 (311-380) authors try to meet the comment.

Point 6: Some specific comments:

Lines 89-171: this whole paragraph is partially out of topic, please consider synthesize this part and maintain only the information necessary in the context of the present review.

Response 6: Part of this paragraph has been omitted while the necessary information has been maintained and synthesized into Subsection 3.1. Wetlands, reed beds and transboundary water situation in Kazakhstan in Lines 192-227 (141-174) and Subsections 5.1. Reed resources development and utilization in the past in Lines 648-678 (313-343) and 5.2. Current development of reed resources and their utilization in Lines 679-696 (344-358).

Point 7: Line 194: in some areas around the world (e.g. North America and Oceania), Phragmites is an invasive species

Response 7: The recommended add has been inserted. Its new location is in Line 168 (126).

Point 8: Table 1: explain the difference between the two types of data reported here, i.e. harvested biomass and total yield. Is total yield the product of reed bed area per harvested biomass?

Response 8: Authors answer is yes. However, due to the comment of another Reviewer, the wordings in Table 1 has been slightly corrected. In Lines 187-188 (136-137): Biomass in Table 1 is expressed in t/ha and the unit containing year as a time factor is productivity given as “Harvested biomass” which was inherited from the source (Köbbing et al., 2013). Thereby, to make refinements, the terms “Harvested biomass” and "Total yield" have been renamed to "Average productivity" and "Total biomass" correspondingly.

Point 9: Tables 1 and 2: what is the difference between the term “harvested biomass” (used in Table 1) and “biomass productivity” (used in Table 2)? The unit of measure is the same (i.e. t ha-1 yr-1).

Response 9:  Authors apologize for this confusing wordings between different tables. The answer: there is no difference. The clarification regarding the terms used in Table 1 is given in previous Response 8.

Point 10: From line 220: section 4 contains a lot of detailed information about the reed beds areas distribution in different provinces of Kazakh SSR. I think this whole paragraph may be scarcely appealing for an international audience

Response 10: During the major revision, authors have removed section 4 with a lot of local information.

Point 11: Table 22: the authors compare here the reed area and biomass between the ‘60s and 1990/1993. No more recent data available? What is the present situation?

Response 11: In-Lines 355-388 (212-244): to meet the comment and improve the manuscript authors developed a new Subsection 3.3. A showcase of a historical overview of reed resources in the Ili Delta. Authors have added available information about the current situation for the Ili Delta. The Ili Delta is the only reed bed area where the size and distribution of wetlands and reed bed areas were studied several times between the 1960s and today. No more recent data are available for other red bed areas.

Please see also in the attachment.

Round 2

Reviewer 1 Report

In my opinion, the paper provides good information but due to the content of the manuscript, the title cannot stand  as it is. There is no perspectives in this manuscript desribed. It should be changed to "Historical data on biomass production in Kazakhstan" or something like that.

Author Response

Response to the Reviewer 1 Comments (Round 2)

Point: In my opinion, the paper provides good information but due to the content of the manuscript, the title cannot stand as it is. There is no perspectives in this manuscript desribed. It should be changed to "Historical data on biomass production in Kazakhstan" or something like that. 

Response: The authors express their sincere gratitude to the Reviewer's comment in Round 2 of the Major revision. As for the revised title of the paper, authors have changed it to "Biomass resources of Phragmites australis in Kazakhstan: historical developments, utilization and prospects".

Please see also in the attachment.

Reviewer 2 Report

The Authors submitted for re-revision manuscript “Biomass perspectives of Phragmites australis in

Kazakhstan”. In the first review, Reviewer pointed out,  that submitted manuscript does not fit to the aim and scope of the Journal that why it should be not reviewed. As Reviewer mentioned (in the first review) the work is well organized. The abstract is quite precisely, and gives an idea about subject of the research. The Authors re-edit first part of manuscript and stressed increasing impact of reed recourse on biomass production in Kazahstan as well on the global biomass production.

The Reviewer maintains his opinion, that the topic of manuscript is not a new issue and does not bring anything new to the knowledge. However, after re-edition and responding to the Reviewers comments manuscript better fits to the aim and scope of the Journal.

The Reviewer maintains his opinion, that Manuscript is rather an inventory study or case study than scientific research but raises important issue regarding to search for potentially new biomass sources.

Based on the information presented above,  I would like to change my previously recommendation.

I recommend this work to publication in Resources Journal.

Author Response

Response to the Reviewer 2 Comments (Round 2)

Point: The Authors submitted for re-revision manuscript “Biomass perspectives of Phragmites australis in Kazakhstan”. In the first review, Reviewer pointed out,  that submitted manuscript does not fit to the aim and scope of the Journal that why it should be not reviewed. As Reviewer mentioned (in the first review) the work is well organized. The abstract is quite precisely, and gives an idea about subject of the research. The Authors re-edit first part of manuscript and stressed increasing impact of reed recourse on biomass production in Kazakhstan as well on the global biomass production.

The Reviewer maintains his opinion, that the topic of manuscript is not a new issue and does not bring anything new to the knowledge. However, after re-edition and responding to the Reviewers comments manuscript better fits to the aim and scope of the Journal.

The Reviewer maintains his opinion, that Manuscript is rather an inventory study or case study than scientific research but raises important issue regarding to search for potentially new biomass sources.

Based on the information presented above, I would like to change my previously recommendation.

I recommend this work to publication in Resources Journal. 

Response: The authors express their sincere gratitude to the Reviewer's positive appreciation of the efforts undertaken in Round 1 of the Major revision to increase compliance with the aim and scope of the Resources Journal. Moreover, the Reviewers comments in Round 2 of the Major revision and new positive recommendation to publication this work in Resources Journal were positively perceived by the authors.

Please also see in the attachment.

Reviewer 3 Report

The authors have revised the manuscript deeply according to referees' suggestions.

Some comments:

  • in Table 2 the difference between average productivity and total biomass is still not clear since the two data have the same unit of measure (i.e. t ha-1 yr-1). Please explain.
  •  In their reply to referees' comments, the authors have stated that no current reed resources assessment for Kazakhstan is available. This remains a limitation of the present work that cannot be overcome, thus the authors should highlight it clearly in the text together with the aim of performing a detailed historical overview of reed resources.

Author Response

Response to the Reviewer 3 Comments (Round 2)

The authors have revised the manuscript deeply according to referees' suggestions.

Some comments:

Point 1: in Table 2 the difference between average productivity and total biomass is still not clear since the two data have the same unit of measure (i.e. t ha-1 yr-1). Please explain.

Response 1: The authors thank the Reviewer for such an appropriate comment in Round 2 of the Major revision. Hence, they bring apologizes for the confusion in differentiation between "Average productivity" and "Total biomass" in Table 1 and Table 2 due to the way of expression of their unit of measure. Indeed, there is a small difference in the unit of measure between them. "Average productivity" is expressed in [t ha−1year−1], i.e. the amount of biomass (tons) at a given time (one year) for the unit of the area (one hectare). In comparison "Total biomass" is expressed in [t ha year−1] since it represents a dry weight of biomass (tons) at a given time (one year) for the whole reed bed area (amount of hectare) for each different Provinces in each row of the Tables mentioned above.

However, due to the necessity to bring refinements, the authors decided to change the unit of measure for "Total biomass" from "[t ha year−1]" to [t year−1] based on simple math calculation formula as following "Total biomass [t year−1]" = "Reed beds area [ha]" x "Average productivity [t ha−1year−1]".

Point 2: In their reply to referees' comments, the authors have stated that no current reed resources assessment for Kazakhstan is available. This remains a limitation of the present work that cannot be overcome, thus the authors should highlight it clearly in the text together with the aim of performing a detailed historical overview of reed resources.

Response 2: The authors express their sincere gratitude for this valuable comment. During Round 1 of the Major Revisions, the authors have been trying to highlight a limitation of the present work in Lines: 82-128 (58-88); 838-852 (398-413). However, to meet this comment in Round 2, the authors tried to highlight the limitation of the present work in the text together with the aim of performing a detailed historical overview of reed resources in Lines: 816-823 (379-386).

Please see also in the attachment.

Round 3

Reviewer 3 Report

The authors have modified the text according to the last minor comments, thus the manuscript can now be accepted for publication in Resources.